# The Feasibility of a Text-Messaging Intervention Promoting Physical Activity in Shift Workers: A Process Evaluation

**DOI:** 10.3390/ijerph20043260

**Published:** 2023-02-13

**Authors:** Malebogo Monnaatsie, Stuart J. H. Biddle, Tracy Kolbe-Alexander

**Affiliations:** 1School of Health and Medical Sciences and Centre for Health Research, University of Southern Queensland, Ipswich, QLD 4305, Australia; 2Department of Sport Science, Faculty of Education, University of Botswana, Gaborone 0022, Botswana; 3Centre for Health Research, Physically Active Lifestyles, University of Southern Queensland, Springfield, QLD 4300, Australia; 4Faculty of Sport & Health Sciences, University of Jyväskylä, 40014 Jyväskylä, Finland; 5Research Centre for Health through Physical Activity, Lifestyle and Sport (HPALS), Division of Physiological Sciences, Department of Human Biology, Faculty of Health Sciences, University of Cape Town, Cape Town 7700, South Africa

**Keywords:** workplace health promotion, physical activity, process evaluation, RE-AIM, shift work

## Abstract

Workplace health promotion programs (WHPPs) can improve shift workers’ physical activity. The purpose of this paper is to present the process evaluation of a text messaging health promotion intervention for mining shift workers during a 24-day shift cycle. Data collected from intervention participants with a logbook (*n* = 25) throughout the intervention, exit interviews (*n* = 7) and online surveys (*n* = 17) examined the WHPP using the RE-AIM (Reach, Efficacy, Adoption, Implementation and Maintenance) framework. The program reached 66% of workers across three departments, with 15% of participants dropping out. The program showed the potential to be adopted if the recruitment strategies are improved to reach more employees, especially when involving work managers for recruitment. A few changes were made to the program, and participant adherence was high. Facilitators to adopt and implement the health promotion program included the use of text messaging to improve physical activity, feedback on behaviour, and providing incentives. Work-related fatigue was reported as a barrier to implementing the program. Participants reported that they would recommend the program to other workers and use the Mi fitness band to continue monitoring and improving their health behaviour. This study showed that shift workers were optimistic about health promotion. Allowing for long-term evaluation and involving the company management to determine scale-up should be considered for future programs.

## 1. Introduction

Shift work is defined as any work schedule outside the normal work hours of 7 am and 6 pm and is often characterized by fixed or rotating shifts, night work, and early morning or late evening work [1,2]. Industries such as healthcare, manufacturing, and mining depend on 24 h service to meet economic and health care demands. Approximately one-fifth of the global workforce undertakes shift work [3].

When compared to employees who work traditional daytime hours, shift workers are at an increased risk for metabolic syndrome, cardiovascular diseases, cancers, overweight and obesity, type 2 diabetes, and mental health issues [4,5,6]. Insufficient physical activity is one of the leading risk factors for non-communicable diseases [7,8,9]. Shift workers may benefit from physical activity health promotion programs due to the health benefits associated with meeting the physical activity guidelines of 150–300 min of moderate intensity physical activity per week [10]. Despite these benefits, only 41% of shift workers meet physical activity guidelines [11].

The workplace provides an ideal environment for health promotion programs to improve physical activity in employees [12]. Research on workplace health promotion programs (WHPP’s) shows that they may be effective in increasing physical activity and quality of life [13]. A stretching exercise program decreased neck and shoulder pain and quality of life in office workers [14]. Several systematic reviews have demonstrated the efficacy of workplace health promotion interventions at increasing physical activity [12,13,15,16,17]; however, they mainly involved normal daytime workers.

The workplace is complex, with multiple factors and determinants that influence physical activity [18,19]. For shift workers, lack of time and work-related fatigue may explain workers’ low levels of physical activity [20]. According to a study conducted in shift working South African doctors, more steps were reported when working longer shifts, thus resulting in more physical activity at work [21]. The factors influencing the implementation and effectiveness of health promotion programs in both shift and regular day workers need to be understood to improve workers’ health. Considering specific work tasks and ergonomics when planning health promotion programs may be useful. For example, a specific strength training program for welders resulted in improvement in working tasks and reduced the subjective determination of exhaustion [22].

Whereas there are still mixed results from studies on the effectiveness of WHPPs to increase physical activity, most were conducted in high-income countries. For the shift work population, it is recommended that studies be expanded to include workers on a variety of different schedules, with modifications to the strategies used to meet individual needs [23].

The reporting of effectiveness outcomes is essential in intervention studies; however, process evaluations are also necessary to determine whether workplace programs were implemented as planned [24]. There are few process evaluations that have been reported in the literature on WHPPs targeting shift workers [25,26,27]. Process evaluation provides context to the research findings and identifies barriers and enablers for translation into real world settings [28]. The RE-AIM framework provides a structure for the assessment of interventions and includes five dimensions: reach, efficacy, adoption, implementation, and maintenance, and has been applied in multiple interventions including those targeting physical activity [29,30]. Through the RE-AIM components, interventions can be assessed at both individual and organizational levels [31]. *Reach* is the total number of participants available to take part in the program along with the assessment of the representativeness of participants. *Efficacy* refers to the impact of the program on key outcomes including both the positive and negative consequences. *Adoption* is the number and representativeness of participants who adopt the program and barriers to adoption. *Implementation* is the extent to which the intervention was delivered as planned and refers to the fidelity and adaptations of an intervention. *Maintenance* can be considered as the degree to which the intervention has been adopted beyond the end of the study period or the potential to be adopted by participants or within the structure of the workplace [30,32].

Therefore, the aim of this study was to conduct a process evaluation, using the RE-AIM framework, of a workplace health promotion intervention that aimed to increase physical activity in shift workers.

## 2. Materials and Methods

### 2.1. Study Design

A mixed methods study design was used to evaluate a workplace health promotion program that aimed to increase physical activity in shift workers who worked a 24-day shift cycle. The RE-AIM framework guided both the qualitative and quantitative methods to evaluate the health promotion program (Table 1).

### 2.2. Recruitment and Participants

An overview of recruitment and the study design is presented in Figure 1. Eligible participants were sent an information sheet via email and were invited to attend the first meeting at their workplace. The inclusion criteria were: being a full-time shift worker at the chosen mining company, having a previous diagnosis whereby physical activity might pose an adverse risk to health, own a smartphone and be willing to download the mobile application, and being able to communicate in English. The workplace had the shift employees working six days in each period (morning, evening and night) followed by two non-workdays resulting in a 24-day shift cycle. Therefore, the program was conducted in two shift cycles of 24 days each. The researcher obtained department information and shift schedules from participants prior to allocation to the intervention or control group. Allocation to the intervention and control groups allowed participants who started the program with similar shift schedules and in the same department to be allocated to one group to avoid contamination [33]. Ethical approval was obtained from the University of Southern Queensland.

### 2.3. Intervention

The physical activity intervention was designed to increase daily moderate-to-vigorous physical activity (MVPA), defined as a walking cadence of ≥100 steps/min for ≥1 min during a 24-day shift cycle. Baseline measures included an online questionnaire and wearing the activPAL accelerometer device (see Section 2.5) during the first 14 days of the first shift cycle to cover four days in each shift and two non-workdays in the first 24-day shift cycle. During the last five days of the first shift cycle, participants were invited to a one-on-one meeting at their workplace. After completing follow-up measures, participants received the *Mi* fitness band (Xiaomi Mi band 5) as an incentive to monitor and improve their lifestyle behaviors after the program had ended. The Mi fitness band 5 is a wristband that measures several health parameters including heart rate, steps, stress level and sleep. The band can be connected to the companion, Mi Fit application on a smartphone via Bluetooth [34].

#### 2.3.1. Action Planning

During the one-on-one meeting, feedback on physical activity and other outcomes recorded by the activPAL were provided (time spent sitting, standing, and stepping). To guide change in physical activity action planning and specific goal setting, the researcher discussed strategies to improve habitual levels of physical activity following the 5As approach (assess, advise, agree, assist, and arrange) [35]. Implementing the behaviour change techniques, including feedback on physical activity, action planning and specific goal setting, is important, as each offers motivation to change behaviours [36]. The discussions also identified possible barriers to change in physical activity behaviour. Each action planning session was planned for 30–45 min at the workers’ workplace at a time convenient for them.

#### 2.3.2. Text Messages

Participants received individually tailored text messages five days per week designed to increase motivation to undertake physical activity. A total of 15 text messages were sent per person and delivered according to the participant’s shift schedule (day, afternoon, or night). The text messages were designed to fit into the COM-B components of capability, opportunity, and motivation to engage in the physical activity [37], and were based on baseline measures and discussions that came from the action planning. For example, a text message targeted to improve the intention to engage in moderate-intensity physical activity was: ‘A 30-min brisk walk can help increase your moderate physical activity’. An example of social opportunity to increase the behaviour was: ‘Today is a great day to get some physical activity before your afternoon shift. Want to suggest a jog this morning to a family member?’. Participants were requested to not respond to text messages, but to use them as motivation to increase their physical activity.

#### 2.3.3. Ecological Momentary Assessment Survey

Participants received an email link to download the mobile application onto their phones to use it for self-reporting their physical activity. The EMA survey took less than 1 min to complete and was sent daily for the 24-days of the intervention period during the second shift cycle. The survey began with: ‘Did you do any physical activity for 10 min at least one time today?’, and the response options were ‘yes’, ‘no’ and ‘do not know’. The subsequent questions required participants to report the physical activity type, location and time spent on the activity. On the days when participants received text messages, EMA surveys were sent 1 h after the text messages to report if they engaged in any physical activity. EMA assessed the participant’s daily current physical activity [38].

### 2.4. Control Group

The control group wore the activPAL at baseline and follow up for 14 days in the first cycle and second cycle. Generic information related to the benefits of physical activity was provided to the control group participants. They did not participate in the action-planning session and did not receive text messages and EMA prompts, and thus were not offered the program (comparison group). After completing follow-up measures, the control group participants also received the *Mi* fitness band.

### 2.5. Data Collection

#### 2.5.1. Questionnaires

Participants reported their age, sex, smoking status, marital status, weight and height, and the impact of shift work on leisure time activities on a questionnaire used in a previous shift work study [39]. The Global Physical Activity Questionnaire (GPAQ) [40] was adapted to measure physical activity at baseline.

#### 2.5.2. Device-Based Accelerometer Measures

The activPAL (PAL Technologies Ltd., Glasgow, UK) accelerometer was used to assess physical activity. This is a small device worn on the thigh that uses static and dynamic acceleration information to distinguish body posture as sitting/lying, standing and stepping, and stepping cadence [39]. Outcome measures included time spent in a walking cadence of 100 steps/min depicting moderate-to-vigorous physical activity, total minutes of sitting time, lying down, standing, and stepping. A minimum of 5 days of activPAL use for a week, including at least one weekend day, provides a reliable estimate of the assessment outcomes [34].

#### 2.5.3. Researcher Logbook

Process evaluation data in line with the RE-AIM framework were collected during the intervention allowing evaluation throughout the study period. An online exit survey and semi-structured interviews were administered at the end of the program. Participants’ enrolment rates and attendance at all meetings were recorded in a spreadsheet, together with action planning sessions attended, dropouts and the reasons for dropping out (an example is attached as Appendix A). Notes were used to record device wear and non-wear times. To record and report on the 5As during action planning, a printed document with all the sections was used for each participant.

#### 2.5.4. Online Exit Questionnaire

At the end of the program, participants in the intervention group were invited to take part in one-on-one interviews and complete an online exit survey for intervention process evaluation. An online exit survey and semi-structured interviews were administered at the end of the program.

Participants in the intervention group were invited to complete the online exit questionnaire link via email invitation after completing the intervention [41]. Examples of questions included, “Did the program meet your expectations?” with options for answers being ‘yes’, ‘uncertain’, or ‘no’. If they answered ‘no’, they were further asked to report the barriers experienced.

Participants (*n* = 17, 68%) also answered questions about the perceived impact of the intervention on their habitual levels of physical activity. The last component of the questionnaire included open-ended questions whereby participants could report on barriers and facilitators that were not addressed in the previous questions (Appendix A).

#### 2.5.5. Interviews

Intervention participants were invited via email to participate in interviews at the end of the intervention. A semi-structured interview guide using the RE-AIM framework explored each participant’s perception of the intervention (Appendix A). Seven interviews were conducted with participants from the intervention group representing all the departments in the mining company. All interviews were conducted online by the first author using Microsoft Teams and lasted between 20 to 30 min. Permission to record was sought from participants before the interview.

### 2.6. Data Analyses

#### 2.6.1. Sample Size

A power analysis was performed based on similar studies; 27 participants per group were sufficient to have 80% power to detect a difference in time spent walking at a difference of 30 min per day or 1200 steps/day (as measured by activPAL) [42,43].

#### 2.6.2. Quantitative Data

The data from the online exit questionnaire were downloaded and summarised in an Excel table. Descriptive statistics, including frequencies, means and standard deviations, were calculated for the demographic characteristics and overall summaries of the questionnaire components. In order to ascertain differences in demographic characteristics at baseline between the intervention and control groups, we used independent *t*-tests [44]. Statistical significance was defined as *p* < 0.05 and all analyses were conducted using SPSS version 27.

#### 2.6.3. Qualitative Data Analysis

The interview transcripts were uploaded to NVivo 12 [45]. Audio recordings of the interviews were transcribed, and a summary of each interview was sent to the participant so they could review the content. A deductive coding based on the RE-AIM framework (reach, adoption, implementation, and maintenance), was used to group themes and sub-themes [46].

## 3. Results

The characteristics of the intervention and control groups are presented in Table 2. Employees who took part in the text messaging intervention to improve physical activity were aged (39.3; SD ± 5.5) years, predominantly (53%) male and overweight with an average BMI of (26.4; SD ± 5.8) kg/m^2^. There were no significant differences in demographic characteristics between the intervention and control group participants. Most reported their health status as average. At baseline, 56% of participants reported that they are motivated to participate in leisure time physical activity. Seventeen participants (68%) completed the exit online questionnaire after completing the intervention and seven semi-structured interviews were performed (28% of the intervention participants). The findings were reported according to the RE-AIM framework elements of reach, adoption, implementation, and maintenance.

### 3.1. Process Outcome Results

#### 3.1.1. Reach

A variety of recruitment strategies were used to reach the mining employees. We sent emails, distributed flyers to shift workers in the workplace, posted the advertisement with a brief explanation about the study on workplace boards, presented during workers’ meetings and received referrals from other enrolled participants. An overview of participants reached, assessed for eligibility and enrolled is presented in Figure 1.

We contacted 92 employees involved in shift work from all departments in the mining company, including the hospital, and the processing and mining pit departments. Ninety-two employees were screened and assessed for eligibility, and 71 provided consent (77%). Reasons for non-consent include no response (*n* = 8) and not eligible (*n* = 13). Sixty participants (84%) completed baseline measures and were allocated to intervention and control groups equally. Of the 60, 51 completed the follow-up measures (85%), and changing shifts resulted in four participants dropping out. All three mine departments were represented (Table 3). A total of 17 out of 25 participants from the intervention group completed the exit survey (68%), and seven took part in the semi-structured interviews (28%) (Figure 1). The other participants did not respond to calls, text messages or WhatsApp messages.

Some facilitators and barriers with regard to reaching the workers in the company were identified. The common theme that facilitated reaching more employees was the information provided during recruitment about the incentive at the end of the program, hence the theme was ‘incentives motivated’. The theme ‘no work management involvement’ highlighted the participant’s perception about recruitment strategies used without the manager as a barrier for recruiting more participants. Other reasons reported by participants during interviews were prior knowledge about the health benefits of physical activity.

#### 3.1.2. Adoption

All participants in the intervention group completed the baseline measures including the activPAL measures and online questionnaires. The action planning meetings were conducted at the employees’ workplaces. Participants who missed the first action planning session were sent reminders via email and text message until all of the sessions were completed by all of the participants.

The themes relating to adoption were developed from analysing the exit questionnaires and semi-structured interviews. Participants that did not complete the follow up measures reported changed shifts and skin irritations from activPAL wear as reasons for not completing the program. Some participants reported their dislike with the activPAL accelerometer during the interviews even though they completed the measures. The number of the accelerometer wear days emerged as a possible barrier to adopting the intervention because of skin irritations reported, thus the theme “ActivPAL wear skin irritations”. Participants perceived the information about physical activity through the action planning session facilitated adoption of the program according to the exit survey responses. The theme “Awareness of the benefits of physical activity” emerged as a facilitator for intervention adoption. The themes, along with the participants’ quotes, are presented in Table 4. A small number of participants reported that the new information gained from the researcher feedback, particularly about sitting and the benefits of reducing sitting time, was well-received. One of the male employees reported that they are now aware of the health effects of prolonged standing and sitting and that they will now change.

Ten out of 17 participants rated the action planning session as ‘excellent’. Nine participants also rated the text messages as ‘excellent’ as their reasons for adopting the program, and eight rated the EMA app and perceived benefits of the program as ‘excellent’. The summary of exit questionnaires revealed positive responses to intervention components enhancing the adoption of the program (Figure 2).

#### 3.1.3. Implementation

Action planning sessions took 30 min on average per person. Each session involved giving participants feedback on their physical activity data downloaded from the activPAL. The feedback was then followed by the use of the 5As to facilitate the discussion about changing physical activity, assisting with goal setting, and assessing readiness for change. The discussions were recorded on a sheet of paper, and then participants were informed about the follow-up measures at the end of the intervention.

The activPAL accelerometer device wear days ranged from 10–14 days. However, data downloaded showed that three devices did not record data due to device malfunction. All three participants agreed to repeat the measurements with a different device.

The EMA application link was sent to participants via email to be downloaded at the end of the action planning session. Participants (*n* = 20) successfully downloaded the app during the meeting. Five participants who could not download the EMA app due to internet connection difficulties were instructed to download it later once they had a good internet connection. Out of the five, two reported that they were not receiving EMA surveys after downloading the app. They were invited to attend another meeting at their workplace to set up the app with the help of the researcher. However, the EMA surveys were still not received due to unknown reasons, resulting in 23 participants receiving the EMA surveys as planned. Participants were sent one EMA survey per day for 24 days during the intervention. Participants responded to 59% of the EMA surveys. The timing of the EMA surveys and text messages were agreed upon, and a pre-determined time was set during action planning. On average, each participant received 15 text messages. All of the text messages were sent as planned, and 10% of the text messages were responded to.

Themes and participants’ quotes relating to intervention implementation are presented in Table 4 and illustrate the facilitators and barriers identified from the exit questionnaires and interviews conducted. Participants’ perceived work-related factors was one of the main barriers to implementing the program. Work-related fatigue was associated with lack of increasing physical activity during workdays, as participants reported that they were too tired to engage in any physical activity after work. Similarly, some participants reported that working night shifts had a negative impact on their sleep patterns. One the other hand, the text messages forming part of the intervention were well-received, and motivated participants to increase their physical activity. Moreover, the *Mi* fitness band was desired by most participants.

The exit surveys revealed that the intervention met expectations of 16 of 17 participants. Only three reported that they experienced some barriers during the program (Figure 3). However, only one participant indicated on the survey that their change of shifts was a barrier for the implementing of the program.

#### 3.1.4. Maintenance

Long-term outcomes were not assessed in this study, with the exit survey and interviews focused on perceptions of the future use of the intervention components. The majority of participants had shown appreciation for the usefulness of the health promotion program. The theme “Continuation use of information from program” emerged to indicate that participants may continue to sustain their change in physical activity. To sustain and monitor healthy behaviour beyond the intervention, *Mi* fitness bands were given as incentives. The majority of participants had shown appreciation of the fitness band and reported that they were using it. The interviews also revealed that participants found the program to be useful and that they would recommend it to other shift workers in the future. Table 4 presents themes and quotes from interviews related to the maintenance of the program.

All participants (100%) reported that they would recommend the program to other shift workers in their workplace and that they intended to continue engaging in physical activities (Figure 3).

## 4. Discussion

The main aim of this study was to conduct a process evaluation of a text messaging intervention to increase physical activity in shift workers based at a mining company. The RE-AIM framework was adopted. Our study findings showed that the program was largely implemented as intended, resulting in good uptake and adherence by employees.

### 4.1. Reach

Overall, results indicate that the program reached 66% of shift workers in the company. The proportion of participants signing up for our study was higher than other workplace health promotion programs, with the reach ranging from 23% to 59% [32,47,48]. The proportion of workers enrolled in our study suggested that the strategy of using several recruitment strategies was successful. In a workplace intervention designed to reduce sitting, participants were approached mainly via email, resulting in a 59% participation rate [48] across four companies, with the higher participation in smaller companies [48]. This was supported by the results on recruitment rates in workplace physical activity interventions, showing higher recruitment rates in smaller companies [49].

However, the analysis from interviews showed that the recruitment in our study may have been higher if the program recruitment was driven by the organisation. Barriers reported by participants in the study suggest that not involving company managers may have decreased participation in the program. Furthermore, participants showed the need for clarity regarding whether the researcher has workplace permission and if the program was communicated with the management. Evidence shows that organisational management support during the recruitment for workplace health promotion programs result in enhanced participation [49]. In a study with low recruitment rates, participants reported that the program had not been effectively communicated by the management teams [32]. To improve future study participation rates, it may be important to prioritise management involvement to encourage workers to participate in health promotion programs.

### 4.2. Adoption

There were high ratings for the intervention components, with most participants reporting the benefit in using these components. The text messages were regarded as useful for improving physical activity. This finding was corroborated by previous studies that indicated that text messaging was feasible and had promising effects on physical activity [50,51,52]. Similar to our study, truck drivers participating in an intervention that also included text messages reported that the messages were appropriate and informative [25]. Although Guest and colleagues used Social Cognitive Theory for behaviour change [53], they recommended that the COM-B model would be the best fit to change behaviours for the health intervention for truckers [25]. In our study, text messages were developed using the COM-B model, with capability, opportunity and motivation components [37] playing an important role in the adoption of our physical activity health promotion program. Therefore, interventions should consider including text messages guided by the COM-B in health promotion programs to improve physical activity in shift workers. Furthermore, the use of evidence- and theory-based frameworks to develop the intervention and conduct the process evaluation resulted in the program being desired by workers in the mine more than the previous workplace interventions. Participants reported that the previous workplace wellness programs focused mainly on weight loss with no feedback on physical activity behaviour or much information on physical activity guidelines. The text messaging intervention has the potential to increase physical activity, thus reducing the disease burden in shift workers.

In addition to text messages, action planning appeared to be important for successful adoption and implementation. The majority of participants attended the action planning session, and those who missed it responded after receiving the reminders. Our study was robust in explaining the benefits of physical activity for shift workers’ health and well-being. Our study findings showed that action planning was important in facilitating program adoption, similar to other health promotion programs [54]. The action planning used with other behaviour change techniques has been shown to elicit motivation in participants and to result in increased physical activity [36]. Similar to our study, using multi-approaches rather than unidimensional ones for office workers resulted in improved posture [55]. Even though shift workers in our study seemed to appreciate the information and understand the benefit of changing their physical activity, work factors, including fatigue, may prevent them from being active. With workplaces involving shift work, it will be important to include physical activity during work times. Furthermore, the action planning session might have overcome other potential barriers to adopt the intervention.

### 4.3. Implementation

The intervention was fully delivered, in the correct order, and within the shift cycle (24 days) as planned, and no adverse events were reported. There were some minor changes in the number of activPAL wear days due to skin irritation complaints. However, this was expected due to the length of the shifts and was reported on the information sheets. Previous studies have also reported skin irritation with activPAL wear [56,57]. Therefore, we accepted 11 days of activPAL wear in place of the initial proposed 16 days of four days per shift. Three days in each work shift and two non-workdays were captured with the 11 days of activPAL wear. However, participants who were willing to wear the device for the 14–16 days were still allowed to. This is supported by a previous study that used a continuous five days of activPAL wear, including the weekend. This was seen as acceptable [58], since all shift schedules and two workdays were included.

Our finding concerning the low EMA response rate is comparable to an EMA study in another low-and-middle income country [59]. We set the EMA prompts according to the pre-agreed time from the action planning session. However, some participants later reported a change of shifts, suggesting that unexpected shift changes might have affected EMA response rates. Previous studies have shown that EMA surveys with a smartwatch can have a significantly higher response rate than those using a smartphone application [60]. The use of smartwatches or fitness bands would have been a better choice to self-report physical activity, especially for the mine workers who are not allowed to carry their phones during work hours. However, smartphones were chosen for the current study due to their being more practical and cost-effective than smartwatches.

However, there were some challenges reported during the program. A few participants reported that the questionnaires were too long and only completed them after several reminders. Long questionnaires, especially at follow up, seem to present a concern for the workers. Thus, shorter surveys with the use of mobile EMA with more ecological validity could be useful to measure healthy behaviours for baseline and follow-up measures. Walthouwer et al. also reported decreased uptake because participants reported that it took too long to complete the questionnaires [61].

### 4.4. Maintenance

Participants were optimistic and reported that they would recommend the health promotion program to other shift workers. Whilst enthusiasm and feedback were positive about continuing with the program, maintenance at the organisational level was not assessed. Therefore, interventions need to evaluate the company setting, especially targeting management to influence intervention uptake and continuity in the workplace [48]. The *Mi* fitness band received as an incentive was useful beyond the period of the intervention. Therefore, giving fitness bands as an incentive rather than money may increase changes in physical activity behaviour.

The results of our study suggest that the text-messaging health promotion program was well received by shift workers. These findings are encouraging, as adapting the intervention to participants’ shift cycles might increase uptake and encourage them to increase and maintain habitual levels of physical activity. In addition, it was encouraging to know that the program was well-received by participants, and that they were all willing to promote it to their colleagues. The text messages in particular encouraged participants to increase their physical activity. Furthermore, workers emphasized the *Mi* fitness band as being a strong motivator for taking part in the program and continued to use it after the program. Workers expressed that they desired the physical activity program, however there is a need to address some work factors including fatigue and more involvement from company management. Our findings therefore suggest that future interventions among shift workers should incorporate action planning based on the shift worker’s schedule, include text messages, and include the use of physical activity monitors/wearables, such as the Mi fitness wristband. Future research should explore the effectiveness of the intervention over more than 1 shift cycle and the optimal frequency and content of the text messages. For workplaces with fatigue-related issues, frequent breaks should be emphasized in future studies to reduce fatigue in order to promote physical activity during workdays.

## 5. Strengths and Limitations

The process evaluation combined quantitative and qualitative components, and therefore provided a strong insight into participants’ perceptions of the health promotion program and also lessons for future research involving shift workers. Furthermore, the intervention development was guided by the behaviour change wheel COM-B components and was evaluated using the RE-AIM framework. However, some limitations should be noted. This study involved one diamond mining company in Botswana. The mining company chosen represents a dominant private employer in Botswana and has similar operations in other towns. The results may not be generalizable to other mines, industries, and countries. Similarly, the workplace in the mining town in the south of Botswana may differ from other mining towns within the same company. Furthermore, we may not have identified all the barriers and potential enablers for increasing physical activity and promoting healthy behaviours. The number of workers recruited did not match our sample size calculation; however, this was a pilot study, and thus 25 individuals per group would be acceptable [62].

## 6. Conclusions

This study showed that a text messaging health promotion program to improve physical activity is feasible for mining shift workers. The health promotion program was well-received, but recruitment strategies involving work management should be considered to increase uptake. Our findings showed that the program has the potential to change physical activity and increase awareness of the benefits of physical activity for the shift work population. Providing feedback on physical activity and giving the fitness band as an incentive were successful approaches that are viable for future workplace health promotions for shift workers. However, understanding the long-term impact of the intervention is needed, including how continued participation by employees and companies may impact the intervention. This evaluation provides insights for researchers and practitioners planning and implementing health promotion programs in a mining workplace targeting the shift working population.

## Figures and Tables

**Figure 1 ijerph-20-03260-f001:**
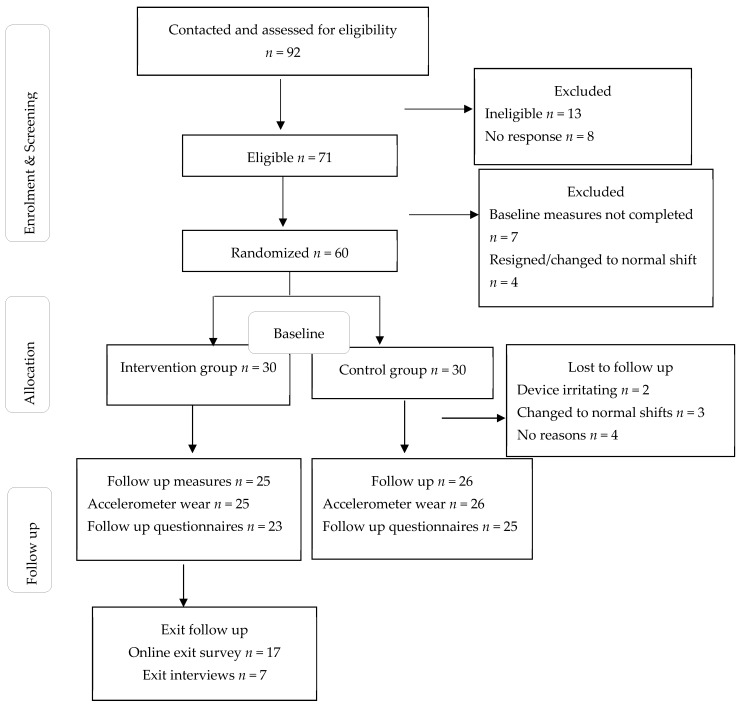
Flow chart of participants.

**Figure 2 ijerph-20-03260-f002:**
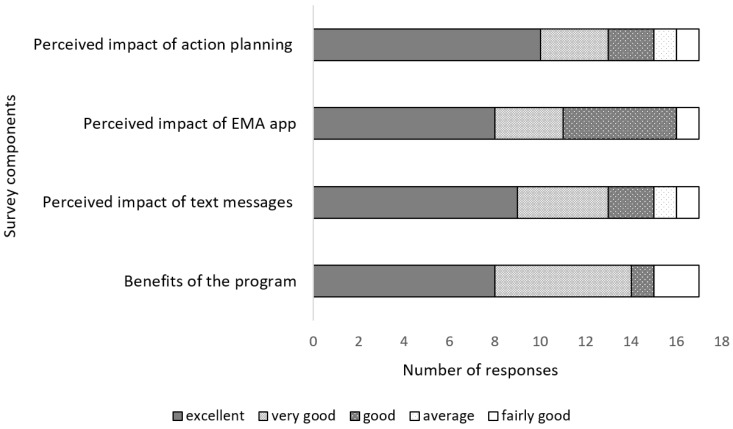
Summary of exit survey questions on adoption of intervention components. **Perceived impact of action planning**: participants reporting how they rate action planning, **Perceived impact of EMA app**: participants reporting their rating for the EMA app, **Perceived impact of text messages:** participants reporting their rating for the text messages, **Benefits of the program**: participants reporting their rating for the health promotion program.

**Figure 3 ijerph-20-03260-f003:**
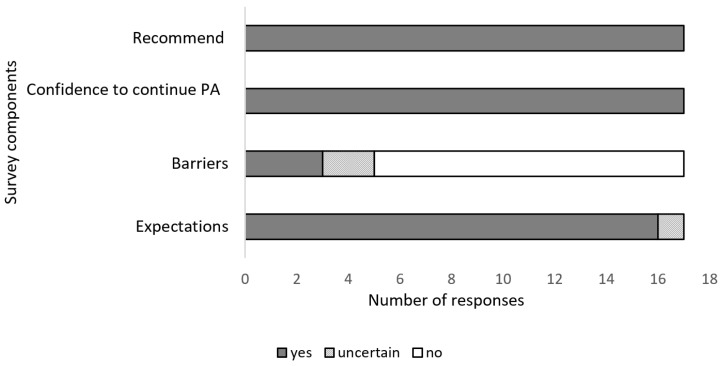
Summary of exit questionnaires on implementation and maintenance of the program. **Recommend**: number of participants reporting that they would recommend the program to other shift workers, **Confidence to continue PA**: number of participants who reported that they will continue to be physically active, **Barriers:** participants reporting that they encountered barriers to taking part in the program, **Expectations**: participants reporting whether their expectations about the program were met.

**Table 1 ijerph-20-03260-t001:** Process evaluation measures and data sources using the RE-AIM framework.

Dimension	Indicator	Data Source
Reach	Participation rate	Researcher logbook
	Dropout rate	Recorded number of enrolled and completed assessment workers
	Individual reasons for non-participation	Researcher logbook and online questionnaires and interviews
	Barriers and facilitators for participation	Online questionnaires and interviews
Adoption	Representativeness of work departments and drop-out	Researcher logbook
	Factors that affect individual participation and engagement with intervention components	Online questionnaires and interviews
	Method used to target various departments	Researcher logbook
Implementation	Barriers and facilitators of intervention process	Online questionnaires and interviews
	Expectations of intervention components	Researcher logbook and online questionnaires
Maintenance	Individual reporting on the continuation of intervention beyond the intervention period	Online questionnaires and interviews
	Barriers to maintaining the program	Online questionnaires and interviews

**Table 2 ijerph-20-03260-t002:** Baseline characteristics of participants in the intervention and control groups [mean, *n* (%)].

Characteristics	Total (*n* = 51)	Intervention Group (*n* = 25)	Control Group (*n* = 26)	*p*-Value
Age years (mean, SD)	39.3 (5.4)	39.1 (5.8)	39.4 (5.2)	0.89
BMI kg/m^2^ (mean, SD)	26.2 (5.8)	26.0 (7.2)	26.4 (4.2)	0.42
Gender (*n*, %)				0.39
Male	27.0 (53.0)	12.0 (44)	15.0 (55.5)	
Female	24.0 (47.0)	13.0 (54.2)	11.0 (45.8)	
Marital status (*n*, %)				0.06
Living with partner	20.0 (41.0)	12.0 (27.1)	8 (17.0)	
Health status (*n*, %)				0.07
Poor	15.0 (30.6)	10 (20.8)	5 (10.4)	
Average	20.0 (40.8)	10 (20.8)	10 (20.8)	
Excellent	2.0 (4.1)	0 (0)	2 (4.2)	
Number of participants reporting LTPA (*n*, %)	30.0 (56)	16.0 (29)	14.0 (27)	0.77

Legend: SD: Standard Deviation, BMI: Body Mass Index computed from weight and height, LTPA: Leisure Time Physical Activity. Differences between the groups were determined via an independent *t*-test with significance *p* < 0.05. Note: Living with a partner included both participants married and those not married living with partners.

**Table 3 ijerph-20-03260-t003:** Employees reached at the company.

Department	Total Number of Employees Who Signed Up for the Intervention	Total Dropout Rate
Hospital mine	30	6
Mining processing	18	2
Mining pit	12	1

**Table 4 ijerph-20-03260-t004:** Themes related to indicators of reach, adoption, implementation and maintenance.

RE-AIM Components	Theme	Facilitator /Barrier	Quotes
Reach	Incentive promised	Facilitator	Female hospital nurse: “*Well, I saw one on the colleagues wearing the fitness band, honestly, I got excited and felt you know, I really need to get this myself. Remember I even approached you and asked you to give me one before I can even join program*”.
	Aware of the benefits	Facilitator	Male pit mine worker: “*Once I heard the program was about wellness, I joined because I am some who likes being active. But I haven’t exerciced since COVID. I thought this is the chance to improve my physical activity*”.
	No work management involvement	Barrier	Male mine processing worker: “*I think because you are a researcher from outside. People in the mine are usually keen if the employer was more involved. Maybe in future the mine management should be involved so more people will participate than just you alone*”.
	ActivPAL wear skin irritations	Barrier	Female hospital nurse: “*I wish there could be something else to use than the stickers because some skin are very sensitive causing some irritation*”.
Implementation	Text messages useful	Facilitator	Male mine pit worker: “*The text messages were helpful. They were a reminder especially on a lazy day and I receive a text. Then I will get up and something. They assisted me to be active*”.
	Program beneficial	Facilitator	Male hospital nurse: “*This program is very reliable to shift workers because they can do their physical activity in a planned manner and time*”.
	Work factors	Barrier	Female hospital nurse: “*I wanted to do more, but because of work, I get home really tired*”.
Maintenance	Continuation in use of information from program	Facilitator	Male mine processing worker: “*The program has helped with the monitoring, so it motivated me to be active. These days I hardly skip gym. Had a slow week last week, but back to gym this week*”.
	Continuation use of the *Mi* fitness band	Facilitator	Male mine pit worker: “*I set myself a standard that it should be 10,000 steps a day so if I haven’t done those 10,000 steps so I stand up and start doing some exercises. Even if it’s not walking now. I have a machine step that I use*.”
	Recommendation to use the program for the company	Facilitator	Female hospital nurse: “*It was a very beneficial program for me personally and for the company. I have been regularly engaging in physical activity ever since participating in the study. I also hope the results will reach relevant authorities*.”

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
