# Peer review of "The Feasibility of a Text-Messaging Intervention Promoting Physical Activity in Shift Workers: A Process Evaluation"

_ijerph, 2023, doi:10.3390/ijerph20043260_

Round 1

Reviewer 1 Report

Promoting activities among shift workers is an important and much discussed topic. 

The authors have presented a scientific process evaluation.

The introduction is close to the state of current research and sufficiently well written. The scientific question is concisely elaborated for the reader at the end.

The description of the methods is very detailed and in view of the results evaluated, the authors should streamline the text here and break up some small sub-items. Line 95 the first mention of the figure should be close to the figure so that the reader does not have to search for it over several pages.

The results are very short in relation to the detail of the methodology. Especially the interview results seem to be very short in the evaluation and presentation.

The discussion is easy to read and the reader can follow the argumentation. The weaknesses and limitations are presented in a separate section and discussed accordingly. the authors could consider excluding this from the discussion.

Overall, the authors tend to write in great detail. there may be potential here to shorten the text for the benefit of the reader and to elaborate the key points more. this probably results from the fact that the work is part of the PhD of the main author.

Author Response

Thank you for the time and your edits in our manuscript, we have addressed each of the comments in file attached.

Reviewer 2 Report

This paper is focused on effect of text messaging program for health promotion. The paper is generally properly written, however, previously it would need revision. The following issues need some address:
1. The literature review is adequate. Several articles published on effects of WP in several countries need to be reviewed for a better overview. Also, it is important not only to describe and explain the issue of work-related problems but also need to link it with the situations that occur during the work. These recently published WP articles need to be reviewed and cited (https://doi.org/10.1080/10803548.2015.1029290, https://doi.org/10.1080/10803548.2021.2014090, https://doi.org/10.1177/0269215515575747).
3. The size of the sample is good but the sampling must be described much more clearly. Why was this choice made? The figure 1 is not displayed properly for showing the drop-offs and other wordings.
4. How the questionnaire was developed? Neither justification of various items nor any supporting literature. Please provide proper justification with questionnaire.
5. The results are of some interest but the authors could elaborate further on the practical implications of their findings.
6. The discussion section needs more information particularly in relation to the theory/model of work-related problems with regard to the findings of the study.

Author Response

(The authors gave the same response as above.)

Round 2

Reviewer 2 Report

Thanks for making changes. However, I have few comments for improvement.
1. What is the implications of this findings? Like what you are suggesting to the users (change in work space, changes in strategies of working, etc.). With this thought, I suggest authors to review the following articles (https://doi.org/10.2486/indhealth.2020-0188, https://doi.org/10.1080/10803548.2021.2014090, https://doi.org/10.1007/s00420-018-1336-1). It will help to suggest future scope of work.

Author Response

Thank you for the feedback. We have added some edits as recommended in the documents attached.
